# Prediction of Potential Suitable Distribution Areas for Northeastern China Salamander (*Hynobius leechii*) in Northeastern China

**DOI:** 10.3390/ani14213046

**Published:** 2024-10-22

**Authors:** Lei Han, Minghang Zhou, Ting Zhang, Wenge Zhao, Peng Liu

**Affiliations:** 1College of Life Science and Technology, Harbin Normal University, Harbin 150025, China; hsdhanlei@163.com (L.H.); 13351886399@163.com (T.Z.); zhaowenge311@126.com (W.Z.); 2School of Mathematical Sciences, Harbin Normal University, Harbin 150025, China; zmh001027@126.com; 3Key Laboratory of Biodiversity of Aquatic Organisms, Harbin Normal University, Harbin 150025, China

**Keywords:** *Hynobius leechii*, MaxEnt, climate change, distribution pattern, species conservation

## Abstract

**Simple Summary:**

In the context of future global climate change, investigating the potential suitable habitats for species is essential for animal conservation, particularly for the most environmentally sensitive amphibians. The Northeastern China Salamander (*Hynobius leechii*), a member of the Hynobiidae family, is endemic to Northeastern China and is classified as a national Class II protected species. This study utilized distribution records of the Northeastern China Salamander to predict its suitable habitat distribution under both current and future climate conditions. The results indicated that several climate variables significantly influence habitat suitability, including precipitation during the warmest season, precipitation in the driest quarter, the seasonal variation coefficient of temperature, average temperature in the wettest season, and mean diurnal range. Currently, the Northeastern China Salamander is mainly distributed in the Liaodong Peninsula of Liaoning Province, the Changbai Mountain area of Jilin Province and a few areas of Heilongjiang Province. Under projected future climate scenarios, the area of suitable habitat is expected to gradually increase, with a continued expansion toward higher latitudes. The findings of this manuscript provide a valuable reference for amphibian conservation efforts in Northeastern China.

**Abstract:**

The Northeastern China Salamander (*Hynobius leechii*) is classified as a rare, nationally protected Class II wild animal in China. Its population is declining, and its habitat is deteriorating. This study aimed to predict the distribution of suitable habitats for the Northeastern China Salamander under both current and future climate scenarios, utilizing the MaxEnt model optimized through ENMeval parameters. Species distribution data were collected from field surveys, existing literature, amphibian records in China, and the Global Biodiversity Information Network. A total of 97 records were compiled, with duplicate records within the ENMTools grid unit removed, ensuring that only one record existed within every 5 km. Ultimately, 58 distinct distribution points for the Northeastern China Salamander were identified. The R software package ‘ENMeval 2.0’ was employed to optimize the feature complexity (FC) and regularization multiplier (RM), and the optimized model was applied to assess the suitable distribution regions for the Northeastern China Salamander under present and future climate conditions. The findings indicated that rainfall and temperature are the primary environmental factors influencing *Hynobius*. Currently, the suitable habitat for the Northeastern China Salamander constitutes 6.6% of the total area of Northeastern China. Projections for the periods of 2050 and 2070 suggest that suitable habitats for the Northeastern China Salamander will continue to expand towards higher latitudes across three climate scenarios. While this study focuses solely on climate change factors and acknowledges certain limitations, it serves as a reliable reference and provides essential information for the distribution and conservation of the Northeastern China Salamander.

## 1. Introduction

In recent years, global climate change has brought new challenges to the survival and protection of wild animals [1]. Relevant studies indicate that over 80% of existing species worldwide will be impacted by rising temperatures and other environmental factors [2,3]. These changes are expected to alter the geographical distribution patterns, species richness, and genetic diversity of wild animals. Amphibians, in comparison to other animals, are ectothermic and exhibit heightened sensitivity to climate change, making them valuable indicator species for monitoring environmental alterations. This increased sensitivity is primarily attributed to their limited mobility and heightened responsiveness to fluctuations in temperature and humidity [4,5,6]. Climate change influences amphibians’ ability to adapt to their habitats, subsequently affecting their geographical distribution patterns and species richness. Notably, among the 93 amphibian species classified as Class I and Class II protected wildlife in China, the Hynobiidae family constitutes 31% [7]. Given these two viewpoints, investigating the geographic distribution patterns and habitat adaptability of Hynobiidae species is essential for the effective conservation of amphibians. Within the context of global climate change, there are two prevailing perspectives regarding the alteration of suitable habitats for amphibians. The first perspective, which is widely accepted, posits that the suitable habitats for amphibians are continuously diminishing. Studies indicate a decline in the number of 662 amphibian species globally [8], with projections suggesting that by 2080, more than half of the world’s amphibian species will experience a loss of over 50% of their existing habitats [9]. In China, it is estimated that amphibian species will lose approximately 20% of their current habitats [10]. Domestic assessments reveal that 43.1% of amphibian species are projected to be threatened in the future [11]. Furthermore, international research on the diversity and habitat selection of amphibians and reptiles supports this perspective [12]. The second perspective posits that rising environmental temperatures will create favorable climate conditions for certain amphibian species, thereby increasing the volume of suitable habitats available to them. Domestic research on the distribution patterns of amphibians in the southwest karst geomorphic region, along with studies on the habitat preferences of the Chinese Warty Newt (*Paramesotriton chinensis*) and international research on salamander distributions, supports this viewpoint [13,14,15].

Species distribution modeling is a research methodology that establishes the relationship between species occurrence and environmental variables, enabling the prediction of regional distribution patterns for target species. This approach has been extensively applied in various fields, including the planning of nature protection areas, regional predictions of alien species encroachment, and the identification of potential distribution areas for key species in the context of ecological restoration [16]. Current studies indicate that the MaxEnt model offers several advantages, including superior fitting of prediction results to actual conditions, the largest predictable area, the most stable outputs, and the capability to effectively validate prediction outcomes [17,18]. Notably, this model demonstrates improved predictive performance even with limited species distribution data [19]. Despite its shortcomings, including low computational efficiency, a risk of overfitting, and dependence on environmental variables [20,21], the maximum entropy model has gained widespread acceptance in habitat suitability analyses and in predicting potential habitats for rare flora and fauna [22,23,24,25], particularly in forecasting the potential distribution of amphibians [26,27,28].

Currently, foreign scholars have utilized the maximum entropy model to generate comprehensive predictions regarding amphibians in various regions, including the Northeastern China Salamander [29]. Research on habitat adaptability and distribution prediction of amphibians utilizing the maximum entropy model in China has primarily concentrated on species such as Chinese giant salamander (*Andrias davidianus*), Wushan Salamander (*Liua shihi*), Jinfo Salamander (*Pseudohynobius jinfo*) and Wenxian Knobby Salamander (*Tylototriton wenxianensis*) [30,31,32]. Additionally, studies have investigated the suitable habitat distribution areas for Hainan Stream Treefrog (*Buergeria oxycephala*), Hanlui Brown Frog (*Rana hanluica*), and Hainan Odorous Frog (*Odorrana hainanensi5*) within the order Anura [33,34,35]. However, there remains a paucity of research concerning the habitat adaptability and distribution predictions of Hynobiidae species in China, particularly regarding the habitat adaptability of amphibians in the unique environmental and climatic conditions of Northeastern China.

The Northeastern China Salamander, belonging to the genus *Hynobius* within the amphibian family Hynobiidae, predominantly inhabits hilly regions at altitudes ranging from 100 to 800 m [36]. This species enters a period of dormancy from early April until late September or early October each year. Its preferred habitat consists of mountain gullies characterized by dense tree coverage, abundant vegetation, and a consistent supply of high-quality water throughout the year, the Northeastern China Salamander is widely distributed across Northeastern China and the Korean Peninsula and is classified as a national Class II protected species. In 2023, it was designated as a vulnerable (VU) species in the Vertebrates Section of China’s Red List of Biodiversity [7,33]. Its conservation is threatened by several factors, including habitat loss, climate change, ecological competition, the impact of pathogens, the invasion of alien species, and pollution resulting from human activities. Notably, it is the only representative of the genus *Hynobius* within the Hynobiidae family found in Northeastern China, highlighting its ecological significance [37].

The Northeastern China Salamander, a rare amphibian, holds significant ecological and social value, warranting focused study and conservation efforts. First, it is a vital component of biodiversity; conserving this species is essential for maintaining the diversity and stability of its ecosystem. Second, due to its sensitivity to climatic and environmental changes, the Northeastern China Salamander serves as an indicator species, with fluctuations in its population reflecting the overall health of the ecological environment. Third, with an evolutionary history spanning approximately 230 million years, this salamander is crucial for research in biological evolution, phylogeny, and species adaptation. Fourth, amphibians play a critical role in ecosystems by controlling pest populations and facilitating material circulation; thus, protecting the Northeastern China Salamander is integral to sustaining these ecosystem services [37,38]. Fifth, as a unique species, it harbors distinctive genetic resources, which may hold potential value for future biotechnological applications. Sixth, research on the Northeastern China Salamander encompasses several critical fields, including ecology, climate change, habitat management, and disease monitoring. Given the effects of climate change and human activities, the urgency for conservation efforts is growing. Through various research initiatives and collaborative efforts, we are dedicated to providing a scientific foundation and practical guidance for the protection of the Northeastern China Salamander, ensuring its continued vital role within the ecosystem. Lastly, conservation initiatives for the Northeastern China Salamander often encompass habitat preservation, benefiting many other wildlife species and indirectly safeguarding broader biomes. In light of global climate change, studying the habitat suitability of the Northeastern China Salamander in its native region is crucial for guiding the conservation of uropods and other ectothermic animals in Northeastern China, thereby establishing a theoretical foundation for ecological protection and sustainable development in the area.

## 2. Materials and Methods

### 2.1. Study Area

Northeastern China comprises the provinces of Heilongjiang, Jilin, Liaoning, and the eastern part of Inner Mongolia, covering an area of 1.46 million km^2^, which represents 15.2% of the total landmass of China. This area is positioned on the eastern fringe of Eurasia, bordered to the east by Korea, to the south by the Bohai Sea, to the west by Mongolia, and to the north by Russia. The region is vast and features a diverse and intricate natural and geographical landscape, including mountains, hills, plains, and wetlands [39]. The eastern limits are defined by the Yalu and Tumen Rivers, while the southern boundaries are outlined by the Yellow Sea and the Bohai Sea. The western border is established by land frontiers, and the northern limit is marked by the Ussuri River and the Heilongjiang River. Within this area, one can observe the high, middle, and low elevations of the Greater Khingan Mountains, Lesser Khingan Mountains, and Changbai Mountains. At its core lies the extensive Songliao Great Plain and the Bohai Depression. Northeastern China extends across the middle temperate and cold temperate zones from south to north, displaying a temperate monsoon climate with four distinctive seasons. Average annual precipitation ranges from about 1000 mm in the southeast to below 300 mm in the northwest. This region supports a wide variety of animal and plant species, making the conservation and sustainable management of these resources essential for promoting local economic growth and preserving ecological equilibrium [40].

### 2.2. Distribution Data Collection and Processing

This research made use of information gathered from the second National Survey of Terrestrial Wildlife Resources, which was carried out by the National Forestry and Grassland Administration, along with a special survey focused on biodiversity conservation conducted by the Ministry of Environmental Protection. The study included maps and distribution data for amphibians in China sourced from Amphibia China and the Global Biodiversity Information Network (http://www.gbif.org/). Only those distribution records with accurate latitude and longitude coordinates were chosen for the analysis. Initially, 97 distribution locations for the Northeastern China Salamander were gathered. The field investigation will be conducted from 2014 to 2022, primarily employing the transect method. This study will encompass a total of 19 sampling points located in Heilongjiang Province and the Changbai Mountain region. To improve the model’s predictive accuracy, these sites were optimized according to the following criteria: (1) discarding inaccurate and duplicated geographical coordinates; (2) excluding duplicate entries within the highest resolution (2.5 arcmin, approximately 5 km) from the World Climate Database (http://www.worldclim.org/), ensuring that each 5 km area contains only one record of distribution. In the end, 58 unique distribution sites for the Northeastern China Salamander were selected for additional analysis (Figure 1).

### 2.3. Environmental Variables

In this study, we selected 19 climate factors and 3 topographic factors. The current climate data sources span from 1950 to 2000 (available at http://www.worldclim.org/), while future climate data were obtained from the Sixth International Coupled Model Comparison Programme (CMIP6), generated by the BCC-CSM2-MR climate system model developed by the National Climate Center in Beijing. We considered three climate scenarios: SSP 126 (low-level GHG emissions), SSP 245 (medium-level GHG emissions), and SSP 580 (high-level GHG emissions). The topographic factors include a Digital Elevation Model (DEM) with a resolution of 500 m, sourced from the Resources and Environmental Sciences Data Center at the Chinese Academy of Sciences (http://www.Resdc.cn/). Additionally, Elevation (elev), Slope (slo), and Aspect (asp) data were calculated based on the DEM data [38,41]. In species distribution modeling, multicollinearity among environmental variables can lead to model overfitting, thereby reducing accuracy. Therefore, correlation analysis among environmental variables is essential before applying the ecological niche model. To mitigate multicollinearity, ArcGIS 10.8 software was employed to process all environmental factor data, resulting in ASCII format data with a consistent range and a resolution of 2.5 arcminutes. The environmental and distribution data were then input into R v4.0.5, utilizing the ENMTools.pl (https://github.com/danlwarren/ENMTools, accessed on 15 October 2024) to calculate the Pearson correlation coefficient (r) for all environmental factors [42,43]. If the correlation between any two environmental variables exceeded ±0.8, we selected the variable with greater biological significance for inclusion in the model (Figure 2) [23,44]. Adjust the geographical coordinate system of the environmental factors to WGS 1984 UTM Zone 48 N. Subsequently, extract the Northeast China region from the environmental factor’s raster data layer. Finally, utilize the ArcToolbox tool to convert the environmental factors raster data and save it in the file format required by MaxEnt. Based on the results from the test runs of all MaxEnt variables, we excluded environmental variables that contributed minimally to the model, ultimately identifying 10 climate variables for modeling (Table 1).

### 2.4. Parameter Optimization and Model Construction

The R software package ENMeval is employed to optimize the MaxEnt model, which incorporates five parameters: linear feature (L), quadratic feature (Q), fragmentation feature (H), product feature (P), and threshold feature (T). The feature combinations analyzed include “L, LQ, H, LQH, LQHP, LQHTP”. The frequency doubling control setting is established within an interval of 0.5 to 6, with increments of 0.5. The optimal model is identified when the deltaAICc value reaches its minimum. Both distribution data (in CSV format) and environmental variables (in ASCII format) are imported into MaxEnt v3.4.1 software. The testing dataset comprises a random selection of 30% of the distribution data, while the training dataset consists of the remaining 70%. The analysis involves checking the jackknife option, setting the number of replicates to 10, selecting Bootstrap for the replicated run type, and retaining the default settings for other parameters. This model employs the knife-cutting method to assess the contribution rates and importance of environmental variables in relation to the distribution of dominant species [45].

In this research, the receiver operating characteristic curve (ROC curve) is utilized to evaluate the model’s accuracy [46]. Currently, ROC curve analysis is viewed as the premier method for assessing viable regions. The area beneath the curve, referred to as the Area Under the Curve (AUC), is determined by the space enclosed by the curve and the *X*-axis, with typical values falling between 0.5 and 1. An AUC value surpassing 0.8 indicates that the model possesses a high level of precision and is considered to have a notable reference value. In contrast, an AUC value exceeding 0.9 implies that the model demonstrates very high accuracy and holds considerable reference significance. Moreover, AUCdiff, which indicates the deviation between the training AUC and the test AUC, is used to assess the extent of overfitting within the model. AUCdiff values that approach 0 denote a lesser degree of overfitting and suggest enhanced predictive performance [47]. In addition, the contribution rates of climate factors obtained from the MaxEnt model, the importance of substitutions, and results from the knife-cutting technique were thoroughly analyzed to determine the key climate factors affecting the distribution of the Northeastern China Salamander. Based on the response curves of the main climatic variables, the suitable ranges for each climate factor impacting the Northeastern China Salamander were established.

### 2.5. Analysis of the Change in Potential Suitable Growth Area

To investigate the changes in spatial patterns of the potential habitat for the Northeastern China Salamander in the future, we utilized ArcGIS 10.8 software to reclassify all predicted results, converting them into a binary matrix file consisting of ‘0’ and ‘1’ based on a Cloglog threshold of ≥0.4 [23]. Specifically, areas categorized as medium and high suitability were identified as potential habitats for the Northeastern China Salamander [41]. Using the contemporary potential habitat area as a baseline, we calculated the future changes in potential habitat areas. The results of this comparative analysis were subsequently processed in ArcGIS 10.8 to visually represent the anticipated alterations in the spatial patterns of potential habitats for the Northeastern China Salamander. All classifications of suitable areas and regional variations were computed using ArcGIS 10.8. The simulated results were categorized according to the habitat suitability index value in *.asc format, with values ranging from 0 to 1, indicating that suitability increases with higher values [45]. Employing the natural break point classification method in ArcGIS 10.8, we divided the predicted suitable habitats for the Northeastern China Salamander into four categories: unsuitable area (0–0.2), low-suitable area (0.2–0.4), moderately suitable area (0.4–0.6), and highly suitable area (0.6–1) [48]. The moderate and highly suitable areas were designated as suitable habitats for the Northeastern China Salamander, allowing us to determine the predicted range of suitable habitats in Northeastern China. Further analyses and forecasts of the trends in habitat changes for the Northeastern China Salamander will be conducted in the future.

## 3. Results

### 3.1. Model Optimization and Accuracy Evaluation

Utilizing data from 58 distribution sites and 10 climate factors, the MaxENT model, optimized through Enmeval data software, was employed to predict the potential distribution areas of *Hynobius* both currently and in the future. When the feature class (FC) is set to LQHP and the regularization multiplier (RM) is 4.5 and the delta AICc value is 0, indicating that the model is optimal (Figure 3). Under these parameters, the prediction results from MaxEnt revealed an AUC value of 0.955 ± 0.008 for the receiver operating characteristic curve (Figure 4), signifying excellent predictive performance.

### 3.2. The Importance of Environmental Variables

As illustrated in Table 1, the MaxEnt model predicts that the cumulative contribution of the top four climate factors accounts for 95.6% of the model’s explanatory power. These factors include Precipitation of Warmest Quarter (bio18, 52%), Precipitation of Driest Quarter (bio17, 26.6%), Temperature Seasonality (bio4, 9.6%), and Mean Temperature of Wettest Quarter (bio8, 7.4%). Furthermore, the proportion of these top four climate factors in the displacement importance value—indicative of the model’s dependence on these variables—reaches 99.3%. The values for these four factors are as follows: Precipitation of Warmest Quarter (bio18, 48.5%), Precipitation of Driest Quarter (bio17, 21.4%), Temperature Seasonality (bio4, 21.4%), and Mean Temperature of Wettest Quarter (bio8, 14.6%). According to the knife test analysis, the top four climate factors ranked by their training gain scores are Precipitation of Warmest Quarter (bio18), Precipitation of Driest Quarter (bio17), Temperature Seasonality (bio4), and Mean Diurnal Range (bio2), as depicted in (Figure 5). A comprehensive analysis of the test results from the three methods (the Percentage Contribution, the Permutation Importance, the jackknife test analysis) indicates that the primary climatic factors influencing the potentially suitable distribution of the Northeastern China Salamander are the Precipitation of Warmest Quarter (bio18), Precipitation of Driest Quarter (bio17), Temperature Seasonality (bio4), Mean Temperature of Wettest Quarter (bio8), and Mean Diurnal Range (bio2). These findings suggest that the distribution of the Northeastern China Salamander is significantly influenced by temperature and precipitation. Based on the response curves of the major climate factors (Figure 6), the optimal ranges for the Precipitation of Warmest Quarter (bio18), Precipitation of Driest Quarter (bio17), Temperature Seasonality (bio4), Mean Temperature of Wettest Quarter (bio8), and Mean Diurnal Range (bio2) for the Northeastern China Salamander are as follows: >494.09 mm, >27.22 mm, <1240.8, >22.04 °C, <10.99 °C.

### 3.3. Distribution and Area of Suitable Habitat for Northeastern China Salamander in Northeastern China Under Modern Climate Conditions

The results of the MaxEnt model (Figure 7) indicate that the suitable habitats for the Northeastern China Salamander are primarily located in the southeastern region of Northeastern China, specifically within the Liaodong Peninsula of Liaoning Province and the Changbai Mountain area of Jilin Province. Under current climate conditions, the total suitable distribution area for *Hynobius leechii* is approximately 96,500 km^2^, which represents 6.6% of the total area of Northeastern China. The limited habitat area presents significant challenges for the conservation of the Northeastern China Salamander. Among this, the high suitability area covers about 47,500 km^2^, predominantly found in the coastal regions of Liaoning Province and near the Yalu River in Jilin Province. The medium suitability area, which encompasses the eastern part of Liaoning Province and the Changbai Mountain region in southeastern Jilin Province, is distributed in a ring-like pattern, covering 48,900 km^2^, or 50.67% of the total suitable habitat. The low suitability area surrounds both the medium and high suitability regions and also includes a scattered distribution in the eastern part of Heilongjiang Province, totaling 73,300 km^2^, which accounts for 5.02% of the total area of Northeastern China. Conversely, the area deemed unsuitable for habitation spans 1,290,000 km^2^, representing 88.3% of the total area of Northeastern China.

### 3.4. Potential Distribution and Changes in Northeastern China Salamander in Northeastern China Under Future Climate Scenarios

The concentration of greenhouse gas emissions significantly influences the distribution of suitable habitats for the Northeastern China Salamander. Between 2050 and 2070, under three distinct climatic scenarios, the potential suitable distribution area for this species is projected to gradually increase, while the unsuitable distribution area is expected to decrease (Figure 8). Notably, the suitable habitat is anticipated to expand towards higher latitudes, with the primary growth concentrated in the eastern regions of Northeastern China. As illustrated in the accompanying table, the total potential distribution area for the Northeastern China Salamander is expected to rise under the three climate scenarios when compared to current conditions in 2050. Specifically, under the SSP126 climate scenario, the area designated as most suitable and sub-suitable is projected to increase by 177,670 km^2^ and 182,730 km^2^, respectively. Under the SSP245 climate scenario, the increases are estimated at 239,180 km^2^ for the most suitable area and 177,080 km^2^ for the sub-suitable area. Finally, under the SSP585 climate scenario, the increases are projected to be 283,380 km^2^ for the most suitable area and 202,690 km^2^ for the sub-suitable area (Table 2).

As illustrated in the table, the total potential distribution area for the Northeastern China Salamander is projected to increase under three climate scenarios by the year 2070, compared to current climate conditions, while the unsuitable area is expected to decrease. Specifically, under the SSP126 climate scenario, the area designated as most suitable and sub-suitable is projected to increase by 205.92 × 10^4^ km^2^ and 206.92 × 10^4^ km^2^, respectively. Under the SSP245 climate scenario, these areas are expected to expand by 350.03 × 10^4^ km^2^ and 145.48 × 10^4^ km^2^, respectively. Lastly, under the SSP585 climate scenario, the most suitable area is anticipated to increase by 382.10 × 10^4^ km^2^, while the sub-suitable area is projected to rise by 138.32 × 10^4^ km^2^.

The area of suitable habitats is projected to increase across three climate scenarios in the upcoming two periods, although notable differences exist among them. In the SSP126 scenario, the total area increase is the lowest, whereas the SSP585 scenario exhibits the highest total area increase. By 2050, the areas of suitable habitats under the SSP245 and SSP585 scenarios increase by the same amount; however, low and medium suitability areas experience greater increases in 2050, while the most suitable areas see the largest increase by 2070. A comparison of the changes in suitable habitat areas between the three scenarios for 2050 and 2070 reveals that the area of the most suitable zone is consistently increasing. In the SSP126 climate scenario, the area of the sub-suitable zone increases, while it decreases in the other two scenarios. Additionally, the area of the low-suitable zone increases under the SSP126 scenario but decreases under the SSP245 scenario, with a slight decrease observed under the SSP585 scenario. The increase in suitable habitats may be attributed to the unique geographical location of Northeastern China. Concurrently, employing models to predict distribution areas can enhance our understanding of future trends in suitable habitats and support the long-term conservation of the Northeastern China Salamander.

## 4. Discussion

In this research, we utilized an enhanced MaxEnt model to forecast the distribution patterns of suitable habitats for the Northeastern China Salamander under both existing and anticipated climate scenarios. The model’s high AUC value demonstrates its strong predictive accuracy [43]. Conversely, numerous previous studies have utilized default parameters when configuring the MaxEnt model [49,50], which may result in overfitting and a significant omission rate, ultimately producing inaccurate predictions. The ENMeval software is commonly employed to refine the regularization multiplier (RM) and feature combination (FC) within the MaxEnt framework, thus improving the accuracy of predictions made by the model [37]. Recently, a growing number of researchers have contended that optimizing the MaxEnt model is essential before making predictions, given that default parameters can introduce particular biases [51]. Our results reinforce this viewpoint.

As poikilotherms, amphibians’ survival and distribution are significantly influenced by external environmental factors. Humidity, temperature, vegetation cover and disease factors in the environment will all affect the distribution of tailed amphibians, particularly changes in climatic conditions [52,53,54]. In this study, the Precipitation of the Warmest Quarter (bio18) and the Precipitation of the Driest Quarter (bio17) emerge as the two most critical climate variables determining the distribution of potential suitable habitats for the Northeastern China Salamander. The distribution of most amphibians is greatly affected by Elevation. Due to the relatively small altitude range in Northeastern China, the average altitude is about 200 m, and the highest altitude is only about 2650 m. However, the Northeastern China Salamander is mainly distributed in hills and mountains with an altitude of 100–800 m. Therefore, in this study, the altitude has little influence on the distribution of the Northeastern China Salamander. Humidity is essential for amphibians, as it significantly impacts their survival, reproduction, and overall health. This finding aligns with the broader understanding that, like many amphibians, the Northeastern China Salamander’s habitat distribution is closely linked to water availability, corroborating previous studies that state, “species richness of amphibians is more strongly correlated with water” [55,56]. From a physiological standpoint, this correlation is understandable, as amphibians possess permeable skin and depend on moist environments for survival. Juveniles typically inhabit aquatic environments, while adults primarily reside on land, with some species remaining in water; however, adults generally prefer wet and cool habitats. Thus, it is evident that the distribution pattern of these amphibians is predominantly driven by their biological needs, rather than indirect factors, reinforcing the notion that water is the primary determinant affecting amphibian distribution [13]. Surface water resources in Northeastern China are limited, and water scarcity significantly impacts the survival and reproduction of amphibians. The warmest season in this region occurs from June to August, during which heavy rainfall can create many temporary pools. These pools can maintain stable water levels for a certain period, providing a more suitable environment for *Hynobius* tadpoles. This helps prevent tadpole mortality due to high temperatures and insufficient water. In Northeastern China, the dry season primarily occurs in winter, and an increase in precipitation during the driest quarter can lead to greater water accumulation. This accumulation ensures adequate humidity for the hibernation of the Northeastern China Salamander and offers more habitat options post-hibernation, aligning with research findings on the habitat environments of amphibians in southwest karst regions [13]. Furthermore, the seasonal variation coefficient of air temperature (bio4), the average temperature of the wettest quarter (bio8), and the Mean Diurnal Range (bio2) influence the distribution of the Northeastern China Salamander. For amphibians, environmental temperature plays a crucial role in their behavior, as their body temperature is largely dependent on external conditions. Fluctuations in temperature can affect behaviors such as evading predators, foraging, and mating [57]. In summary, the primary climatic factors influencing the distribution of *Hynobius* are related to water and temperature, corroborating the conclusion that these two factors are critical in determining the distribution of amphibians. Our research findings indicate that the most suitable habitats for the Northeastern China Salamander are primarily located in the coastal regions of Liaoning Province and the border region of Jilin Province. This suggests that the species may have specific ecological preferences in these locales [58]. Analyzing the key climatic factors that influence the distribution of the Northeastern China Salamander, we conclude that these areas are in close proximity to bodies of water, such as the sea or rivers, which contribute to higher humidity levels conducive to the species’ survival. However, this concentrated distribution also renders the salamander more susceptible to threats posed by climate change and regional natural disasters, thereby increasing the risk of extinction for the entire population [59].

In the future, the suitable habitat area for the Northeastern China Salamander is projected to continue expanding. By the years 2050 and 2070, the largest suitable habitat area is expected under the SSP585 scenario, followed by SSP456, with SSP126 showing the smallest area. This finding aligns with the perspective of Wang and colleagues, who suggest that higher concentrations of greenhouse gas emissions generally result in a broader range of climatic factors, such as precipitation and temperature, thereby leading to a greater change in habitat area [60]. Under scenarios characterized by high emissions and carbon usage, the suitable habitat area for the northeast small salamander is anticipated to expand further. This conclusion is consistent with studies on other salamander species, including the Blue-spotted Salamander (*Ambystoma laterale*), Red-Back Salamander (*Plethodon cinereus*), and Wushanba Salamander (*Hynobius yiwuensis*), which also indicate that suitable activity ranges increase with rising temperatures and greenhouse gas concentrations [32,61]. Domestic research has demonstrated that both tailed and tailless amphibians are likely to expand their suitable habitats in the future. For instance, a study in the Hengduanshan area found that *Batrachuperus pinchonii*, a member of the family Liliidae, exhibits the most significant increase in suitable habitat area across three emission scenarios [62]. The future niche for amphibians in karst and non-karst landforms is projected to increase by 24.1% and 14.2%, respectively [13]. Similarly, foreign studies have demonstrated that the suitable habitat area for certain amphibian species is expected to expand significantly in the future. In the Mexican rainforest, for example, the suitable habitat area for 22% of tailless amphibians is projected to increase by more than 50% [63]. The suitable habitats for salamanders in the western United States are expanding [15]. The changes in the suitable habitat area for the northeast small salamander align with these findings. We hypothesize that these results are linked to the northeast small salamander’s origin in the colder regions of the northeast. This species is now finding suitable habitats in the southeastern part of Northeastern China, characterized by higher temperatures and humidity, and this trend is expected to continue as temperatures and rainfall increase in future climate scenarios. Consequently, the temperature and humidity levels in other colder areas of the northeast have reached conditions conducive to the northeast small salamander’s survival, leading to a continued northward expansion of its suitable living area. The data indicates that the northeast small salamander is persistently spreading northward into higher altitudes. These findings are consistent with the distribution patterns of amphibians in the Hengduan Mountains and salamanders in Canada and the northeastern United States. This supports the conclusion that amphibians generally tend to migrate towards higher elevations and latitudes [62,64]. Additionally, the research corroborates the observations made by Yao and colleagues regarding the significant northward and eastward distribution trends of Lepidopodidae species, with little evidence of southward movement [65]. The results of this study provide a comprehensive prediction of the future diffusion trends of suitable habitats for the northeast small salamander and offer valuable insights for its research and conservation.

Methods that leverage climate data and potential changes in key factors influencing species survival to assess species distributions may exhibit certain limitations [15]. In some cases, alternative approaches may surpass the performance of the MaxEnt model [66]. Although the MaxEnt model has been effectively utilized to predict the distribution of both tailed and tailless amphibians [67,68], it is evident that the model is inadequate in quantifying species locomotor capabilities, neglects evolutionary potential, and overlooks biological impacts. This shortcoming leads to the identification of certain existing distribution areas as unsuitable for habitation [23,50]. Consistent with other studies, this research posits that climate is the primary driver of species distribution [69,70] while failing to account for additional factors such as the limited mobility of amphibians, geographical conditions, human interference, infectious diseases, and variations in water availability. While other habitat increases may occur, climate change is widely regarded as a significant threat to global amphibian conservation. Research indicates that amphibian populations worldwide are likely to continue their decline as new threats associated with climate change emerge [71]. Nonetheless, our findings indicate that the MaxEnt model remains a valuable tool for predicting future suitable habitats [72,73]. Collectively, while these factors can enhance our understanding of species’ ecological niches and distributions, their integration poses significant challenges. Modeling that incorporates a broad array of variables and their interactions requires more specific data and may risk overparameterization. Therefore, the development of distribution models is crucial for exploring species distribution and improving our understanding of the effects of individual environmental factors.

## 5. Conclusions

This research utilized the MaxEnt model to assess the potential habitats suitable for Hynobius in both current and future contexts. The results revealed that various climatic factors have a substantial effect on habitat suitability, such as precipitation during the warmest period (bio18), precipitation in the driest quarter (bio17), the seasonal variation coefficient of temperature (bio4), average temperature in the wettest season (bio8), and the mean diurnal temperature range (bio2). At present, optimal habitats are mainly found in the Liaodong Peninsula of Liaoning Province and in the bordering area of Jilin Province in China. Predictions for future climate scenarios indicate a potential increase in suitable regions, with new habitats likely to arise in higher latitudes and suitable distributions also expected in the southeastern section of Heilongjiang Province. These findings contribute to a deeper comprehension of the distribution of the Northeastern China Salamander and offer essential insights for pinpointing its appropriate distribution zones in Northeastern China. This presents a new direction for future research on amphibians in Northeastern China, contributing to the understanding of biodiversity and the conservation of organisms in the region.

## Figures and Tables

**Figure 1 animals-14-03046-f001:**
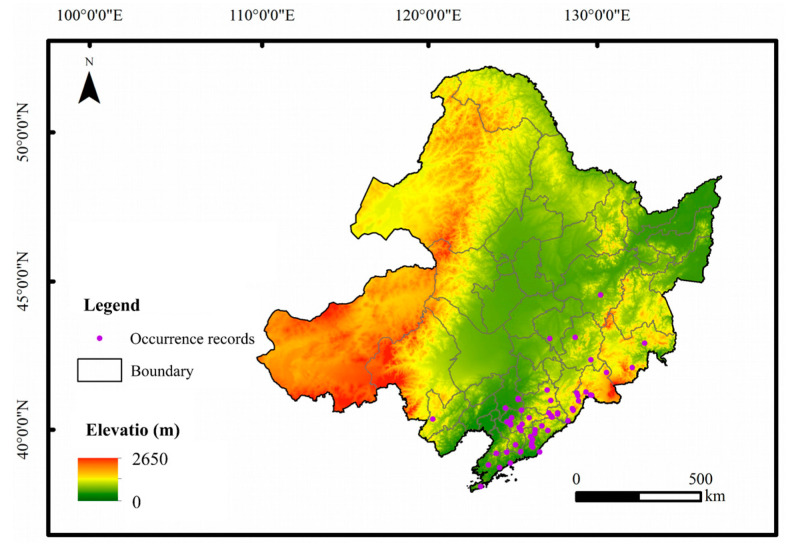
Distribution points of Northeastern China Salamander.

**Figure 2 animals-14-03046-f002:**
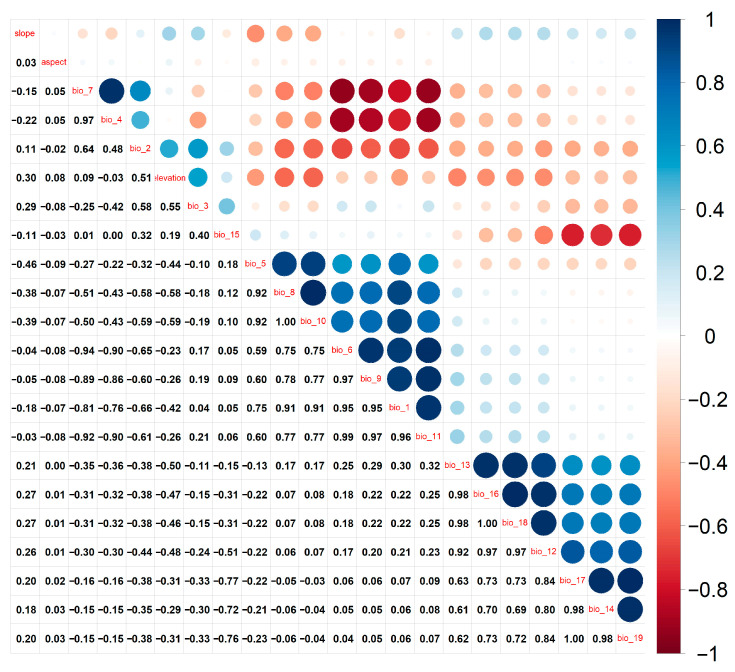
Correlation matrix between environmental variables of Northeastern China Salamander.

**Figure 3 animals-14-03046-f003:**
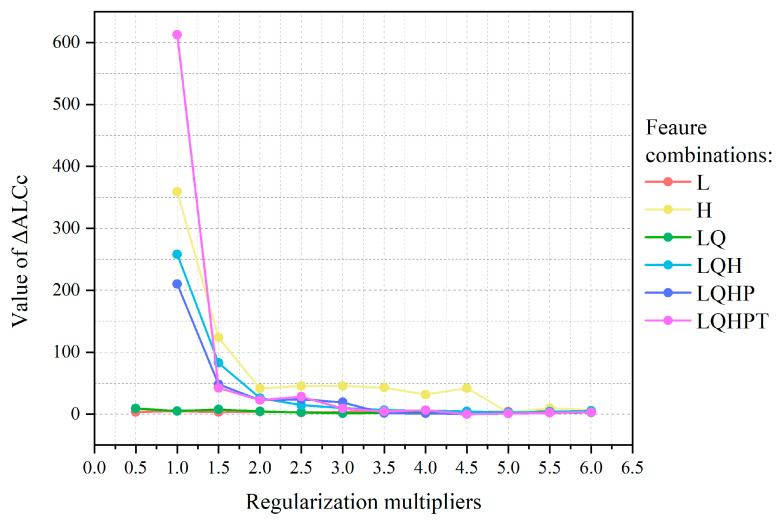
Delta AICc of the MaxEnt models under different regularization multipliers (RMs) and feature combinations (FCs).

**Figure 4 animals-14-03046-f004:**
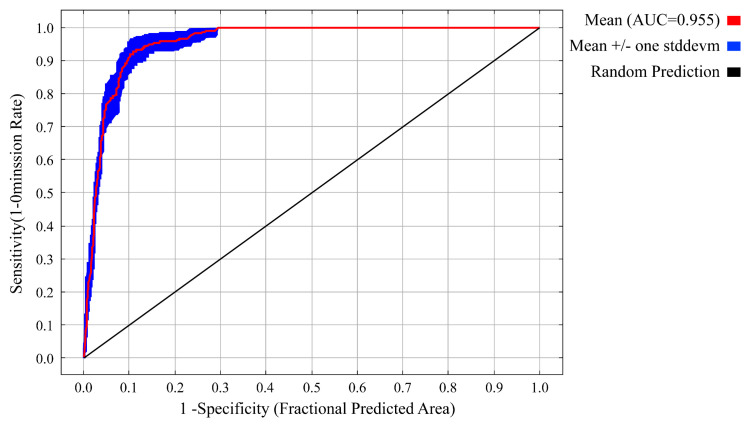
Receiver operating characteristic (ROC) curve and AUC value.

**Figure 5 animals-14-03046-f005:**
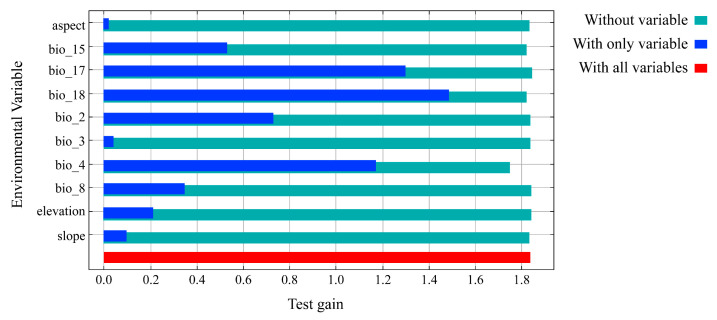
Jackknife test gain for environmental variables in Northeastern China Salamander.

**Figure 6 animals-14-03046-f006:**
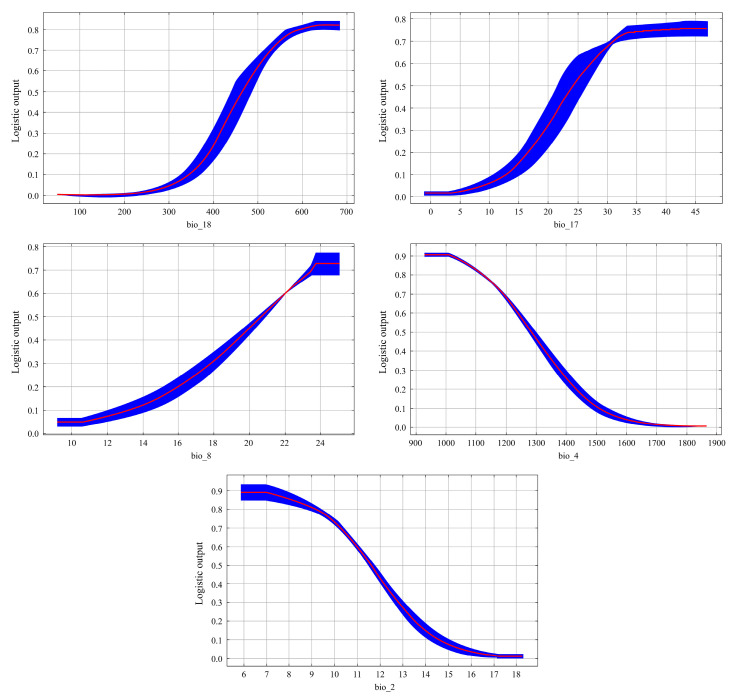
The response curves of major climate factors (Red is the response curve, blue is the standard error).

**Figure 7 animals-14-03046-f007:**
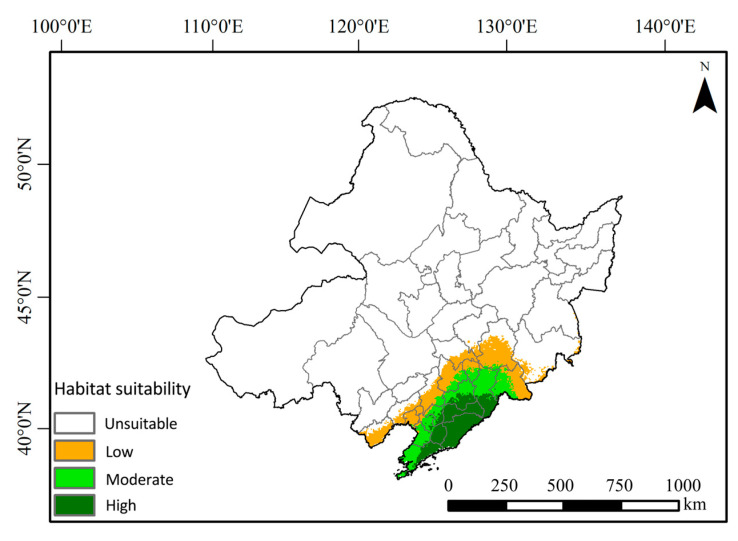
Potential suitable habitats of the Northeastern China Salamander in Northeastern China under current climatic conditions.

**Figure 8 animals-14-03046-f008:**
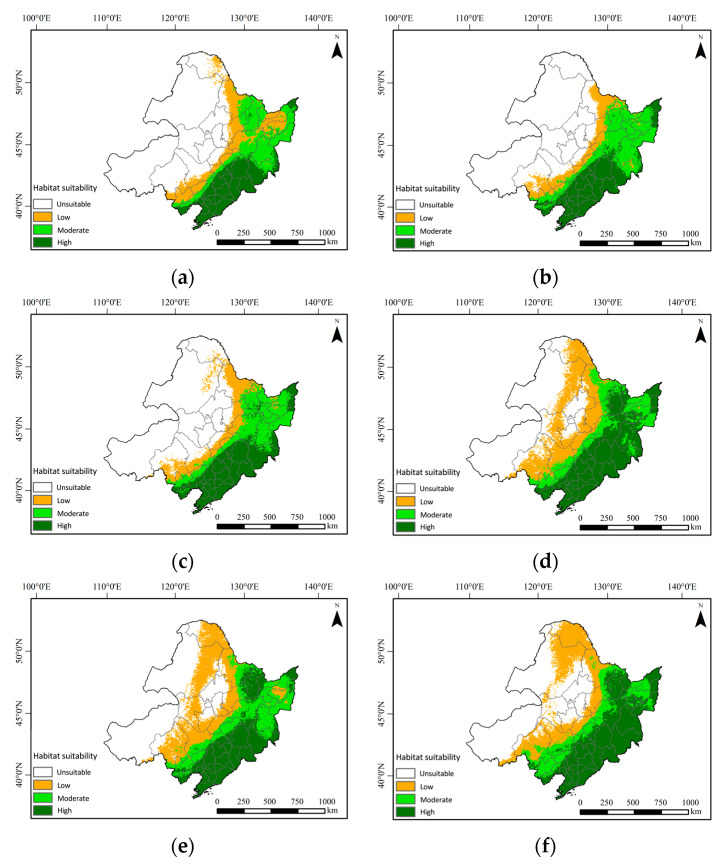
Potentially suitable climatic distribution of Northeastern China Salamander under different climate change scenarios: (**a**) SSP126 in 2050; (**b**) SSP126 in 2070; (**c**) SSP245 in 2050; (**d**) SSP245 in 2070; (**e**) SSP585 in 2050; (**f**) SSP585 in 2070.

**Table 1 animals-14-03046-t001:** Contribution and permutation importance of environmental variables in MaxEnt models.

Code	Environmental Variable	Percentage Contribution (%)	Permutation Importance (%)
bio18	Precipitation of Warmest Quarter	52	48.5
bio17	Precipitation of Driest Quarter	26.6	21.4
bio4	Temperature Seasonality	9.6	14.8
bio8	Mean Temperature of Wettest Quarter	7.4	14.6
bio15	Precipitation Seasonality	2.4	0.1
bio3	Mean Diurnal Range	1.2	0
bio2	Mean Diurnal Range	0.5	0
Asp	Aspect	0.1	0.4
Ele	Elevation	0.1	0
Slo	Slope	0	0.1

**Table 2 animals-14-03046-t002:** The area of suitable habitats across different periods, with values in parentheses indicating variations compared to the current period (×10⁴ km^2^).

Grade	Current	2050			2070		
		SSP126	SSP245	SSP585	SSP126	SSP245	SSP585
Low	7.325	20.011	16.813	34.124	14.083	31.829	30.468
		(12.686)	(9.488)	(26.799)	(6.758)	(24.504)	(23.143)
Moderate	4.900	23.173	22.608	25.169	25.592	19.448	18.732
		(18.273)	(17.708)	(20.269)	(20.692)	(14.548)	(13.832)
High	4.753	22.520	28.671	33.091	25.345	39.756	42.963
		(17.767)	(23.918)	(28.338)	(20.592)	(35.003)	(38.210)
Total	16.978	65.704	68.092	92.384	65.020	91.033	92.163
		(48.726)	(51.114)	(75.406)	(48.042)	(74.055)	(75.185)

## Data Availability

Data are contained within the article.

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
