# Peer review of "Prediction of Potential Suitable Distribution Areas for Northeastern China Salamander (Hynobius leechii) in Northeastern China"

_animals, 2024, doi:10.3390/ani14213046_

Round 1
Reviewer 1 Report
Comments and Suggestions for Authors
Dear editor and authors,
The authors tried to analyze the Impact of Climate Change on the Suitable Distribution of Northeastern China Salamander (Hynobius leechii) in Northeastern China. This research will help the conservation of the salamander.
However, the authors did not provide adequate details of data collection and processing. Original data was not demonstrated well by the authors. And some results (e.g. elevation normally has clear Percentage Contribution) seemed to be unusual but the authors did not discuss or explain.
More detailed comments and suggestions were presented in attached PDF file.
In conclusion, current manuscript did not fully meet the requirements of submission, resubmission after solving problems mentioned is recommended.
Best wishes

Need to be improved.
Reviewer 2 Report
Comments and Suggestions for Authors
The authors present MaxEnt models characterizing current and future habitat availability for the Northeastern China salamander based on climate projections of influential variables. The study is sufficiently conducted and summarized, although the interpretations for salamander populations worldwide might be portrayed with too broad a perspective, suggesting that climate change in general should increase habitat availability. Additional care to regional and life history variation in response to changing climate conditions would be prudent to qualify some of these general interpretations. Habitat loss and degradation is one of the primary contributions to global amphibian declines, and climate change is not the only driver of alterations in habitat. Additional insights into the current conservation status and threats for this rare and declining population, would be more appropriate, recognizing that habitat availability, particularly changes over a 20-year period, would not independently change the conservation status of this species, and the projections for salamanders worldwide could be overstated here. Some specific suggestions follow:
A complete listing of the bio variables would be useful, particularly for interpretation of Figure 1, either in a table and/or figure legend. Variable names in current rendition of this figure are shown in font too small for print.
Line 84 – Consider rephrasing as “species occurrence and environmental variables” to avoid repeating “distribution” in the definition.
Line 86 – “self-heating”? Does this refer to endotherms?
Line 152 – Only two seasons are described following mention of four distinctive seasons.
Line 156 – It seems that reference to section 2.2 should be a heading; however, it is not preceded by section 2 or 2.1.
Line 229 – likewise, section heading 2.2.4 seems out of place and not placed as a heading.
Line 247 – sentence fragment
Line 273 – Which three methods? Those referenced in Figure 3? Reiterate parenthetically or introduce at beginning of paragraph.
Figure 4. – Consider replacing x-axis labels with variable names for easier interpretation.
Line 304 – Does the 73,300 km2 refer to only low suitability habitat? That seems high, given the total suitable habitat area is 96,500 km2 and the medium suitability habitat is half that. Clarify.
Line 428 – Projecting habitat gain as a broad “international” phenomenon is misrepresenting complex and variable impacts of climate change that are dependent on species and location. Revise this section to appropriately portray species and regions with habitat least impacted or expanding as a result of climate change in the appropriate context.
Comments on the Quality of English Language
Some minor grammatical revision and attention to sentence structure is advised.
Reviewer 3 Report
Comments and Suggestions for Authors
The paper presents a detailed analysis of the potential impact of climate change on the distribution of the Northeastern China Salamander (Hynobius leechii), a species of considerable ecological significance. Utilizing the MaxEnt model, the study forecasts the current and future distribution of suitable habitats for this species under different climate scenarios. The authors identify key environmental variables, such as precipitation and temperature, that influence the salamander's habitat and predict an expansion of suitable habitats toward higher latitudes under future climate conditions. These findings offer valuable insights for conservation efforts, particularly in addressing the risks posed by climate change to this vulnerable species.
Your choice of topic is both important and timely, given the escalating pressures of climate change on vulnerable species. However, after a thorough review, I have identified several concerns that may affect the manuscript’s suitability for publication in its current form.
Key Issues:
- Over-reliance on the MaxEnt Model: The study heavily depends on the MaxEnt model for predicting species distribution. While this model is well-regarded, it is prone to overfitting, particularly if not carefully parameterized. Overfitting can lead to predictions that do not generalize well to actual conditions, potentially resulting in inaccurate conclusions. The manuscript briefly mentions model optimization but lacks a detailed discussion on how overfitting was specifically addressed and the implications of the model's limitations. I recommend adding a discussion on these potential limitations, possibly in the methods or discussion section.
- Environmental Variables: The study primarily focuses on climate and topographic variables but does not incorporate other potentially relevant factors, such as land use changes or human activities. Please acknowledge the exclusion of these factors, justify the choice of variables, or suggest how future studies could address this gap.
- Habitat Suitability Analysis: The results section provides a detailed analysis but could benefit from a more explicit connection between the findings and their implications for conservation strategies. Please make sure to link the findings directly to conservation efforts, possibly in the discussion section.
Detailed Line-by-Line Critique and Suggestions for Revisions:
- Title and Abstract:
- Title: The title could be more concise to enhance clarity by removing redundancy.
- Abstract: The abstract effectively summarizes the study but does not mention the key limitations or the need for future research. Consider adding a brief sentence that acknowledges the study’s limitations and the need for further work.
- Introduction:
- Lines 51-82: This paragraph could be shortened for better readability.
- Lines 51-52: The sentence "In recent years, the intensification of human activities, alongside social and economic development, has exacerbated global climate change, presenting new challenges for the survival and protection of wildlife." can be simplified for clarity.
- Lines 80-82: The sentence "Given these two viewpoints, investigating the geographic distribution patterns and habitat adaptability of Hynobiidae species is essential for the effective conservation of amphibians." could be revised to clarify the study's objectives earlier in the introduction.
- Lines 88-95: The manuscript does not address the limitations of using the MaxEnt model. Add a sentence or two that discusses the potential risks of overfitting and the assumption of static species-environment relationships.
- Lines 107-117: The authors should elaborate on the threats faced by the species.
- Lines 118-135: Expand on what previous studies have done and justify how this study differs. The introduction could benefit from an additional review of the target species on a regional or global scale. Please review previous studies and mention the techniques used in studying the target species.
- Materials and Methods:
- Lines 139-157: Please ensure proper citations are included in this paragraph.
- In this section, it is not clear how many background points were used for model building against the presence records. Please clarify this.
- Lines 185-190: The statement "If the correlation between any two environmental variables exceeded ±0.8, we selected the variable with greater biological significance for inclusion in the model." needs justification for excluding non-climatic variables such as land use changes and human activities.
- Consider noting that while AUC is a useful metric, it alone may not be sufficient to evaluate model performance; other metrics like Kappa and TSS are also important.
- Results:
- Line 259-260: The verb in "The analysis involve checking the Jackknife option, setting the number of replicates to 10, selecting Bootstrap for the replicated run type, and retaining the default settings for other parameters." should agree with the singular subject.
- Line 295: The statement "The total suitable distribution area for Hynobius leechii is approximately 96,500 km², which represents 6.6% of the total area of Northeastern China." should explicitly link the findings to conservation strategies.
- Discussion:
- Lines 370-372: The sentence "The Precipitation of the Warmest Quarter (bio18) and the Precipitation of the Driest Quarter (bio17) emerge as the two most critical climate variables determining the distribution of potential suitable habitats for the Northeastern China Salamander." should also discuss the potential for species adaptation and its implications for the model.
- The authors should include references to salamander studies, noting whether similar results have been observed in other salamanders or amphibians. For example:
- Escoriza, D., & Hernandez, A. (2022). Vegetation cover and occurrence of salamanders in the western Mediterranean. Integrative Zoology, 17(3), 456-467.
- Wan, B., et al. (2024). Environmental factors and host sex influence the skin microbiota structure of Hong Kong newt (Paramesotriton hongkongensis) in a coldspot of chytridiomycosis in subtropical East Asia. Integrative Zoology.
- Line 336: The sentence "The area of suitable habitats is projected to increase across three climate scenarios in the upcoming two periods, although notable differences exist among them." should acknowledge geographical limitations and suggest future research directions.
- A small section highlighting the benefits of the applied modeling techniques for establishing priority zones for management actions is necessary. Additionally, mention the limitations of the applied modeling techniques in a few sentences.
- Conclusion:
- Lines 481-482: The statement "These findings contribute to a deeper comprehension of the distribution of the Northeastern China Salamander and offer essential insights for pinpointing its appropriate distribution zones in Northeastern China." should be strengthened by emphasizing the implications for future research and conservation.
- Figures and Tables:
- Figure 5, Line 308: The figure’s caption is brief and lacks detail. Additionally, consider including an inset map in Figure 5 to show the location of Northeastern China within China.
- References:
- While the references appear comprehensive, some may be outdated or less relevant. Please review and update them as necessary.
There are many more formatting errors in the article, which I have highlighted, please see the attachment。
I hope you find this feedback constructive and helpful as you refine your work. Thank you for your contribution to this critical area of research, and I look forward to seeing how you might address these challenges in future revisions.
